# Enhanced rabies surveillance in roadkill specimens by real-time RT-PCR

Crystal M. Gigante[1]*, Claire Hartloge[1], Rene Edgar Condori[1], Jordona D. Kirby[2], Lauren Hovis[1], Kathleen M. Nelson[2], Ryan Wallace[1], Yu Li[1], Richard B. Chipman[2]

1 Poxvirus and Rabies Branch, Division of High-Consequence Pathogens and Pathology, United States of America Centers for Disease Control and Prevention, Atlanta, Georgia, United States of America, 2 USDA, APHIS, Wildlife Services, National Rabies Management Program, Concord, New Hampshire, United States of America

* CGigante@cdc.gov

## Abstract

Roadkill specimens are an important source of samples for enhanced rabies surveillance (ERS) in areas where other methods of sample collection may not be practical. However, the physical condition of roadkill specimens is unpredictable and, in many circumstances, unsatisfactory for rabies diagnostic testing by antigen detection methods. The high sensitivity of real-time reverse transcriptase PCR (RT-PCR) holds promise for rabies diagnostic testing of poor-quality samples. We conducted an evaluation of real-time RT-PCR to detect rabies virus RNA in roadkill samples. A total of 299 specimens were collected from raccoons (n = 232), skunks (n = 46), foxes (n = 17), coyotes (n = 2), a bobcat (n = 1), and a domestic cat (n = 1) across ten states during 2018 – 2021 in the United States. Eight samples (2.7%) were positive using the LN34 pan-lyssavirus real-time RT-PCR assay. These eight rabid animals in areas of high interest for wildlife rabies management would likely not have been identified otherwise. These findings support the use of real-time RT-PCR for samples that would typically be unsuitable for testing by widely used antigenic-based detection methods such as the direct fluorescent antibody test (DFA or FAT) and direct rapid immunohistochemistry test (DRIT).

## Author summary

Roadkills represent potential unrealized samples for detecting and monitoring diseases in wildlife. However, roadkills are often in poor condition, which can affect test results. In this study, a PCR test (LN34 assay) was able to detect rabies virus in eight out of almost 300 poor quality roadkill samples sampled across ten states in the United States. Real-time RT-PCR was successful at detecting rabies virus RNA in poor quality roadkill samples, and results informed rabies surveillance in wildlife in areas of high interest for wildlife rabies management.

**Data availability statement:** All data is included in the publication or available on NCBI databases under GenBank PV807767-PV807770 and PV845156-PV845159.

**Funding:** The author(s) received no specific funding for this work.

**Competing interests:** The authors have declared that no competing interests exist.

These findings support the use of real-time RT-PCR for samples that would be unsuitable for testing by widely used test such as the direct fluorescent antibody test (DFA or FAT) and direct rapid immunohistochemistry test (DRIT).

## Introduction

In the United States, approximately 90,000 animals are tested for rabies each year [1]. Most rabies testing is performed in one of 130 public health laboratories [1]. Rabies public health surveillance testing is critical to ensure that human exposure to a rabid animal is identified promptly to inform decisions about post exposure prophylaxis [2,3]. Public health rabies surveillance can also provide critical data about the distribution and spread of specific rabies virus variants in wildlife when variant typing occurs post-diagnosis, but the data can be biased by human and animal interactions, given sampling depends on reported exposures [4,5]. Particularly for wildlife rabies management, it is also of interest to determine the distribution and prevalence of rabies in wildlife in geographic areas that are not well represented in public health data. Targeted, active surveillance called enhanced rabies surveillance (ERS), is performed to complement and address gaps in resolution from the public health surveillance data. The United States Department of Agriculture, Animal and Plant Health Inspection Service, Wildlife Services, National Rabies Management Program (hereafter USDA-WS) first implemented targeted ERS in 2004 to help inform science-based decision making related to the control of raccoon rabies virus variant (RRVV) [6].

Two regions of ERS in the United States focus (1) along the border of the RRVV enzootic region where oral rabies vaccination (ORV) is used to prevent the westward and northward spread of this variant [4,7] and (2) in Arizona to monitor rabies virus variants in gray foxes, skunks and bats. ERS conducted along and beyond the RRVV ORV zone is critical to inform management, contingency actions, and strategic planning for rabies elimination in mesocarnivores [8]. Surveillance effort in Arizona tends to focus on detection of bat rabies virus spillover into mesocarnivores [9,10], as has been reported previously in the northern part of the state [11–13].

During 2006–2015, USDA-WS observed that rabies positivity rates varied among different ERS sample collection methods, where certain types of animals (e.g., strange-acting) may be more likely to be rabies positive than others (e.g., roadkills) [14]. Since 2016, USDA-WS has implemented a categorization and point system to prioritize the collection of samples with a higher likelihood of rabies detection [6]. The ERS sample categories include 1) reported strange-acting animals, 2) animals found dead not along roads with no obvious explanation, 3) animals found as roadkills, 4) animals trapped and euthanized for rabies surveillance, 5) otherwise healthy reported or nuisance animals, and 6) animals lacking information so categorized as unknown. Roadkills were determined to have the third highest rabies positivity rate after found dead animals and strange-acting animals, and roadkills made up 16% of rabid animals detected through USDA-WS ERS during 2016–2019 [15]. Roadkill sampling can provide some advantages over other ERS sampling methods because roadkill

surveillance can be performed proactively by field biologists and expand the spatial coverage for ERS sampling in areas where other sample types may be difficult to anticipate or acquire systematically.

A major limitation to testing roadkill specimens is the greater likelihood for poor brain sample condition following vehicle collisions. Oftentimes, roadkills may not be identified for several days, leading to further degradation of samples, particularly in warm environments. The antibody-based detection methods, the direct fluorescent antibody (DFA) test and direct rapid immunohistochemical test (DRIT), are highly sensitive to tissue degradation and require brain tissue preserved by cold chain for testing [16–18]. Previous studies have shown that PCR-based assays have higher sensitivity than DFA in poor quality samples and are capable of detecting rabies virus RNA in deteriorated tissues unfit for DFA testing [19–28], suggesting PCR may be an appropriate method for detection of rabies virus RNA from roadkill specimens. The LN34 real-time RT-PCR assay has shown good performance detecting rabies in deteriorated or poor-quality samples [28]. In this study, we evaluated the ability of the LN34 assay to detect rabies virus RNA from roadkill samples deemed unsatisfactory for testing by antibody-based tests collected near the RRVV ORV zone and from Arizona.

## Methods

### Samples

All samples were obtained postmortem from roadkill animals between July 2018 and December 31, 2021; no live animal use or testing was performed as part of this study. Targeted sample size was estimated using Cohran's formula based on expecting a positivity rate of 2% and accepting a 1.5% error rate with 95% confidence (z = 1.96). Roadkill samples were collected in Alabama (AL), Arizona (AZ), Georgia (GA), Maine (ME), North Carolina (NC), New York (NY), Ohio (OH), Tennessee (TN), Virginia (VA) and West Virginia (WV) through opportunistic or formal roadkill surveys during USDA-WS' standard ERS operations but using a Field Protocol for Animal Sample Collection for RT-PCR Testing (see S1 Text). Roadkill surveys were conducted throughout the vaccine bait zones across many states with varying road types and traffic levels. Opportunistic sampling involved biologists picking up roadkill samples of target species often while driving to other work sites or to and from the office. Formal roadkill surveys involved standardized routes driven on a regular schedule. Location, host, animal age, collection date, current weather, photograph, current temperature, previous day temperature, road type, skull condition, brain condition and number of days in the field (S1 Table) were estimated by Wildlife Services biologists or technicians with experience in roadkill sample collection in the region. Sample received, sample description, sample detail, and detailed description of submitted sample were all determined by CDC laboratory staff with rabies testing experience.

In some cases, whole heads, skin, or other sample types were collected (full sample details can be found in S1 Table). AZ was unique from other states because all samples submitted from AZ were animal heads. Serial subsamples were collected from each head at CDC, with an average of 2.8 subsamples tested from each head (ranging from 1 – 7, S1 and S2 Tables). Tissues tested included skull fragments, skin, maggots, and unrecognizable tissue from inside of skull and brain tissue (when present and obvious). Samples were stored at or below -20°C until shipment to CDC. A photograph of the roadkill was taken prior to sample manipulation and collection. The template data sheet filled out for each sample collected can be found in S1 Text. RNA extraction and LN34 real-time RT-PCR was performed at CDC Atlanta as described in [28,29], except PCR was run in duplicate for LN34 and singlicate for internal control β-actin real-time RT-PCR assay (actin assay). This decision was based on data from a multisite study that showing low variability between replicates for the LN34 assay [28].

### Case detection rates

Case detection rates (CDR; percent positivity) in roadkill samples collected during this study were calculated for 266 samples that were located in states where USDA-WS conducts ERS and for which GPS location data were available. CDR from USDA-WS ERS data were generated from results obtained from the National Rabies Surveillance System for

the years 2019 – 2021 and limited to test records that indicated the DRIT test was conducted (NY was excluded because DRIT test was not indicated), states involved in this roadkill study (excluding AZ), and with GPS locations for sample collection (n = 10,582). CDR were then stratified by location relative to the USDA-WS ORV zone (East of the zone, within the zone, and West of the zone). Mid-P Exact p-values were calculated for the comparison of (a) LN34 Poor Quality Samples vs LN34 Good Quality Samples and (b) LN34 All Samples vs DRIT-based ERS. LN34 "Poor Quality Samples" included samples where brain condition was recorded as poor or very poor. LN34 "Good Quality Samples" included samples where brain condition was recorded as fair or good.

## Cost estimate

LN34 cost estimate was based on US dollar pricing listed on the manufacturer's website in the United States during September 2022 for AgPath-ID One-Step RT-PCR Reagents (ThermoFisher 4387391), Direct-zol RNA Miniprep kit (Zymo Research, R2052), and TRIzol Reagent (ThermoFisher 15596018). A cost estimate of $10 per sample for PCR testing was calculated based on 20 samples run in (1) singlicate for LN34 and host internal control β-actin (actin) real-time RT-PCR assays or (2) duplicate in LN34 and singlicate in actin assays with controls on a 96 well plate. An additional $0.50 per sample was added to the estimate for cost of disposable sterile tubes and pipette tips. Cost estimate may not represent what is available at the time of publication, and costs and availability of reagents can vary substantially from location to location. Costs of primers, probes, and positive control RNA were not considered in the cost estimate. Personnel costs were estimated at $20/sample for accessioning, RNA extraction, testing and result interpretation. Field collection and sampling costs were not considered. Additional details can be found in S3 Table.

## Modeling

A multivariable logistic regression model was created in R (version 1.4.1106) to assess conditions associated with samples which had definitive LN34 test results (positive or negative versus indeterminate). The outcome variable was created to reflect samples with indeterminate results (coded as 1) and those with definitive test results (e.g., positive or negative; coded as 0). Arizona was excluded from the model, and only states near the raccoon rabies variant ORV zone were included. Variables measured at the time of sample collection were used as predictors: (1) Skull Condition, (2) Brain Condition, (3) Days in the Field, (4) Previous Day's Temperature, and (5) Current Day's Temperature (S1 Text). Skull condition was categorized by (1) intact, (2) fractured, (3) severely fractured, (4) flattened, (5) no identifiable structures, or (6) other. Brain condition was categorized as (1) good, (2) fair, (3) poor, (4) very poor, or (5) other. Backwards selection was applied in a stepwise process, with variables with the highest p-values removed first until remaining variables had p-value ≤0.05. The AIC and chi square p-value tests were used to assess model fit. The resulting model output is referred to as the sample degradation score and is presented as a continuous value ranging from 0 – 1.

Most of the roadkill samples tested in this study produced indeterminate results by LN34 PCR for either lack of identifiable brain tissue or severe degradation of total RNA causing the assay's internal β-actin control to fail. Indeterminate results do not provide information that can be used in management actions and add to the total cost of roadkill surveillance. We examined the sample data to determine if any field or specimen characteristics correlated with the probability of returning an indeterminate result. Subjective characterization of brain and skull condition in the field was determined by the field biologist. A cost-benefit analysis was conducted based upon the aforementioned model outputs, to compare differing roadkill sampling strategies which would incorporate an LN34 testing scheme. The cost and benefit of collecting poor-condition roadkill samples and testing by LN34 was evaluated across the sample degradation score, at 5% intervals. A roadkill sample-size goal of 1,000 samples was used for the scenario by applying fractional proportions from the observation dataset to a theoretical population of 1,000 roadkill specimens that reflect the same characteristics as the samples encountered during this study. Analysis outputs included the cost per sample tested, cost per positive case detected, and the number of positive cases detected, at 20 intervals of the sample degradation score.

Plots of results and of the U.S. map were made in R using the ggplot2 [30], usmap (https://cran.r-project.org/web/packages/usmap/index.html), dplyr (https://cran.r-project.org/web/packages/dplyr/index.html), and forcats (https://cran.r-project.org/web/packages/forcats/index.html) packages. The ORV zone GIS data was imported using the packages sp (https://cran.r-project.org/web/packages/sp/index.html) and rgdal (https://cran.r-project.org/web/packages/rgdal/index.html). Latitude and longitude of roadkill samples were plotted using plot_usmap. Figures were finished in InkScape (inkscape.org). The U.S. states shapefile was accessed from the Census Bureau (cb_2018_us_state_500k): https://www.census.gov/geographies/mapping-files/time-series/geo/carto-boundary-file.html. The rabies oral vaccination zone shapefile was generated by USDA-WS.

## Sequencing

Rabies virus LN34 amplicon sequencing was performed using Sanger sequencing as described in Condori *et al.* (2020) [31]. Complete rabies virus nucleoprotein and glycoprotein gene sequencing was performed using the Oxford Nanopore MinION as described in Gigante *et al*. (2020) [32]. Sequences generated as part of this study were deposited to NCBI (https://www.ncbi.nlm.nih.gov/) under the GenBank accession numbers PV807767 - PV807770.

## Results

Between July 2018 and December 31, 2021, USDA-WS collected 299 roadkill specimens (in total) from the following states: Alabama (AL), Arizona (AZ), Georgia (GA), Maine (ME), North Carolina (NC), New York (NY), Ohio (OH), Tennessee (TN), Virginia (VA) and West Virginia (WV), along with coordinate location, environmental condition (i.e., weather, temperature), and sample condition (i.e., time in field, brain condition, and skull condition) (S1 Table).

Thirty-two of 299 roadkill samples were submitted for real-time RT-PCR testing from AZ (Fig 1, Table 1). There are distinct rabies virus variants that circulate independently in gray foxes (AZ gray fox variant), striped skunks (south central skunk variant), and multiple species of bats in AZ [1]. Arizona samples were collected from 13 striped skunks (*Mephitis mephitis*, 40.6%), nine gray foxes (*Urocyon cinereoargenteus,* 28.1%), four raccoons (*Procyon lotor*, 12.5%), three hooded skunks (*Mephitis macroura*, 9.4%), one hog-nosed skunk (*Conepatus leuconotus*, 3.1%), one kit fox (*Vulpes macrotis*, 3.1%), and one coyote (*Canis latrans*, 3.1%). None of the samples tested positive by LN34 real-time RT-PCR; nine (28.1%) were rabies negative and 23 (71.9%) were indeterminate due to lack of brain tissue or tissue decomposition.

The remaining samples were collected in or around the known geographic range of RRVV (Fig 1). Most samples were collected from raccoons (228), followed by striped skunks (29), red foxes (*Vulpes vulpes*, 4), gray foxes (3), bobcat (*Lynx rufus*, 1), coyote (1), and domestic cat (*Felis catus*, 1). Eight of the 267 samples were positive (3.0%); 75 were rabies negative (28.1%) and 184 (68.9%) were indeterminate due to lack of brain tissue or tissue decomposition. Positivity ranged from 0% to 10.7% (WV) among the states, though the number of samples submitted also varied from 7 (TN) to 64 (VA), which contributed to variability in percent positivity (Table 1).

Rabies virus RNA was detected from seven raccoons and one striped skunk (Table 2). Six of the eight positive samples were found east of the ORV zone, where RRVV is enzootic, and two cases were from within areas managed with ORV during the study period (Fig 1). The eight rabies positive roadkill samples ranged from good to very poor brain condition and were estimated to have been in the field from <1 to >3 days (Table 2, Fig 2). Five of the eight positive samples were recognized as containing or likely containing brain tissue that varied from good to very poor condition by laboratory testing personnel. However, three out of the eight positive samples contained no recognizable brain tissue and will be described in more detail. The first sample was a small piece of desiccated, leathery brown tissue that produced an LN34 Ct value of 28.6. The second sample was collected from scraping a skull fragment with very little desiccated tissue, maggots, and an oily film; this sample produced a Ct value of 32.5 after 1:10 dilution of the original RNA in water. Initial testing of RNA from this sample produced Ct of 37.2 (indeterminate result). Improvement of Ct value upon dilution of the RNA indicates PCR inhibition possibly caused by the maggots or the oily film. Lastly, a dark red, goopy liquid containing a lot of blood with no

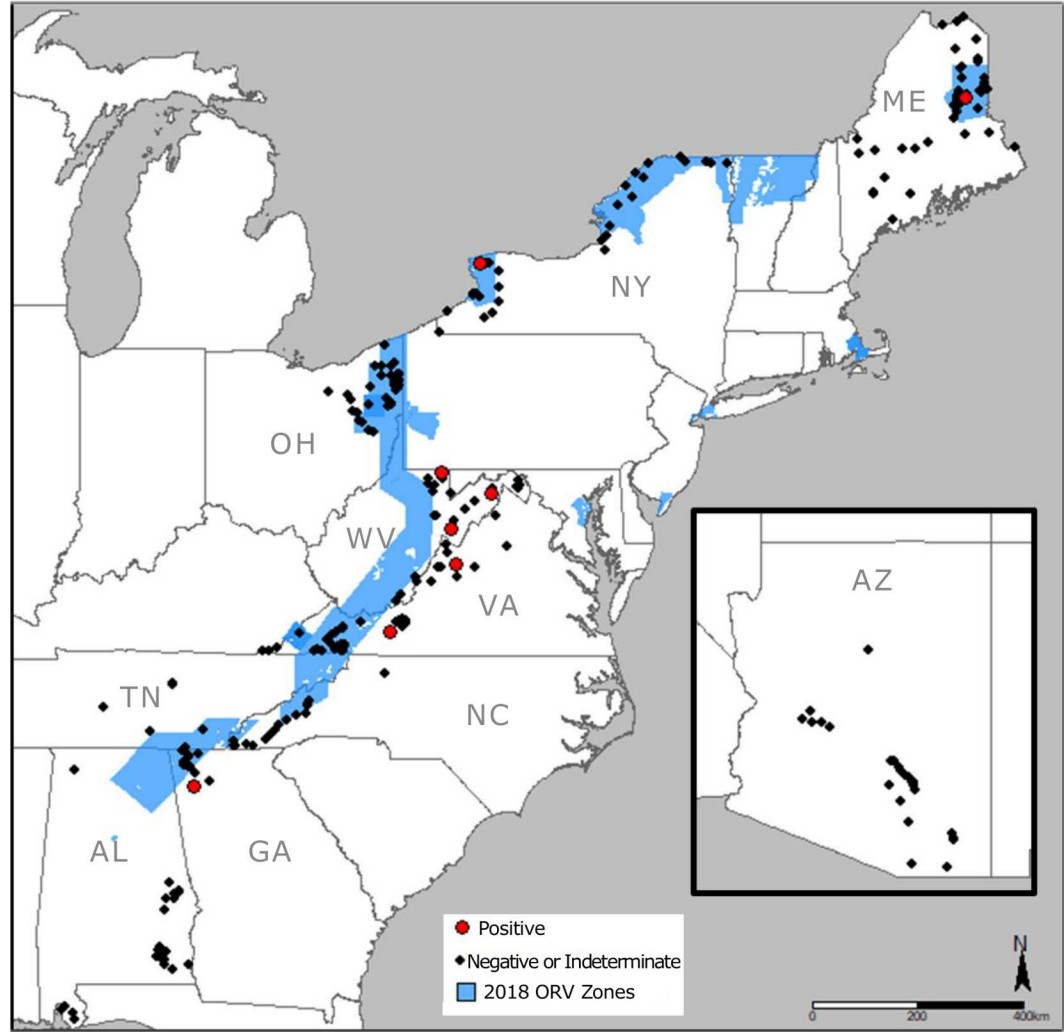

**Fig 1. Location of roadkill samples tested for rabies using the LN34 real-time RT-PCR test (red dots indicate positive results, black dots indicate negative or indeterminate results) relative to the 2018 oral rabies vaccination (ORV) zones (in blue) in the eastern U.S.**

visible brain tissue produced a Ct value of 20.4. Rabies virus variant assignment was possible for seven of the eight positives by sequencing, and they were all RRVV.

Case detection rates (CDR) for roadkill samples collected for LN34 testing showed no significant difference when comparing those that were designated as poor or very poor quality ("LN34-Poor Quality") to those that were either good or fair quality ("LN34-Good Quality")(Mid-P exact p-value range 0.78 – 0.88; Table 3), although very few samples of good or fair quality were tested in this study. Rabies CDR was higher for roadkill samples tested by LN34 when compared to ERS samples tested by DRIT east of the ORV zone (5.0% compared to 1.8%, Mid-P exact $p = 0.03$); however, no difference was observed in CDR in roadkill by LN34 real-time RT-PCR compared to ERS DRIT surveillance within or west of the ORV zone (Mid-P exact p-value range 0.37 – 0.96; Table 3).

LN34 Ct values produced from roadkill samples varied widely, ranging from 13.7 (strong positive) to 38.3 (inconclusive). The recommended Ct value cut-off for rabies diagnosis using LN34 on fresh (unfixed) frozen brain samples is Ct ≤ 35 [28]. Two roadkill samples produced Ct values > 35 and were further investigated to determine if cross contamination may have

**Table 1. Roadkill samples collected by USDA-WS and tested for rabies using the LN34 real-time RT-PCR test; mean percent positivity and 95% confidence interval is shown in the total line.**

**LN34 real-time RT-PCR test**

| State | Tested | Positive | Percent Positive |
|---|---|---|---|
| Alabama | 23 | 0 | 0% |
| Arizona | 32 | 0 | 0% |
| Georgia | 15 | 1 | 6.7% |
| Maine | 48 | 1 | 2.1% |
| North Carolina | 20 | 0 | 0% |
| New York | 30 | 1 | 3.3% |
| Ohio | 32 | 0 | 0% |
| Tennessee | 7 | 0 | 0% |
| Virginia | 64 | 2 | 3.1% |
| West Virginia | 28 | 3 | 10.7% |
| Total | 299 | 8 | 2.7% (1.4-5.2%) |

**Table 2. Details of roadkill samples that tested positive for rabies using the LN34 real-time RT-PCR assay. Presence of lyssavirus RNA is indicated by average LN34 Ct value (LN34) ≤ 35, where lower numbers generally indicate higher RNA levels. Actin Ct values (Actin) indicate presence of host β-actin mRNA and are indicative of sample quality, where high Ct values may indicate poor sample quality or RNA degradation. A full description of how questionnaires were filled out is described in the methods. Temperature (Temp).**

| State | County | Species | LN34 | Actin | Brain Identified | Weather | Temp (°F) | Skull condition | Brain condition | Days in field |
|---|---|---|---|---|---|---|---|---|---|---|
| GA | Floyd | Raccoon | 36.33 | 29.58 | Yes | Sunny | 78 | Intact | Poor | 1-3 |
| ME | Aroostook | Raccoon | 28.62 | 27.78 | No | Rain | 38 | Other | Very poor | Unknown |
| NY | Niagara | Raccoon | 20.45 | 24.61 | No | Sunny | 47 | Fractured | Very poor | >3 |
| VA | Pulaski | Striped Skunk | 32.53 | 23.50 | No | Overcast | 71 | Severely fractured | Very poor | >3 |
| VA | Rockbridge | Raccoon | 13.68 | 19.90 | Yes | Sunny | 61 | Intact | Good | <1 |
| WV | Pendleton | Raccoon | 20.62 | 19.32 | Yes | Sunny | 75 | Severely fractured | Poor | 1-3 |
| WV | Hampshire | Raccoon | 17.10 | 29.63 | Yes | Sunny | 90 | Fractured | Poor | 1-3 days |
| WV | Preston | Raccoon | 17.16 | 23.04 | Yes | Sunny | 70 | Severely fractured | Poor | 1-3 days |

occurred. Each sample was re-extracted and re-tested by PCR multiple times. One sample (raccoon from Floyd County, GA) was extracted four times and LN34 amplification was observed in 18 out of 35 replicate reactions (51.4%) (S4 Table). This sample was processed on the same day as a positive sample; however, gloves and equipment used were all single use and the sample was placed directly into a tube after collection. Contamination was unlikely, and it was determined to be a positive case. The second sample in question, from Niagara County, NY, produced an LN34 Ct value of 38.3. Retesting revealed LN34 amplification in 5 out of 8 replicates. This weak signal was determined to be caused by cross contamination for two reasons. First, sample leaking was observed from the tins containing samples of a batch shipment of roadkill samples from NY (all tins were in one plastic bag, including a strong positive sample). Second, the same re-usable scalpel was used for collection of this sample and a strong positive sample that produced LN34 Ct of 20.5, although the scalpel was disinfected between uses.

The majority (163 of 188, 86.7%) of samples with very poor brain condition produced indeterminate PCR results, while most fair or good samples were determined to be positive or negative (22 of 33, 66.7%; Fig 3). Among samples producing a positive or negative result (n = 92), 25 (27.2%) had very poor brain condition, 42 (45.7%) had poor brain condition, 13 (14.1%) had fair brain condition, 9 (9.8%) had good brain condition, and 3 had unknown brain condition (3.2%, Fig 3). However, it should be noted that three of the eight positive cases had very poor brain condition, so excluding very poor

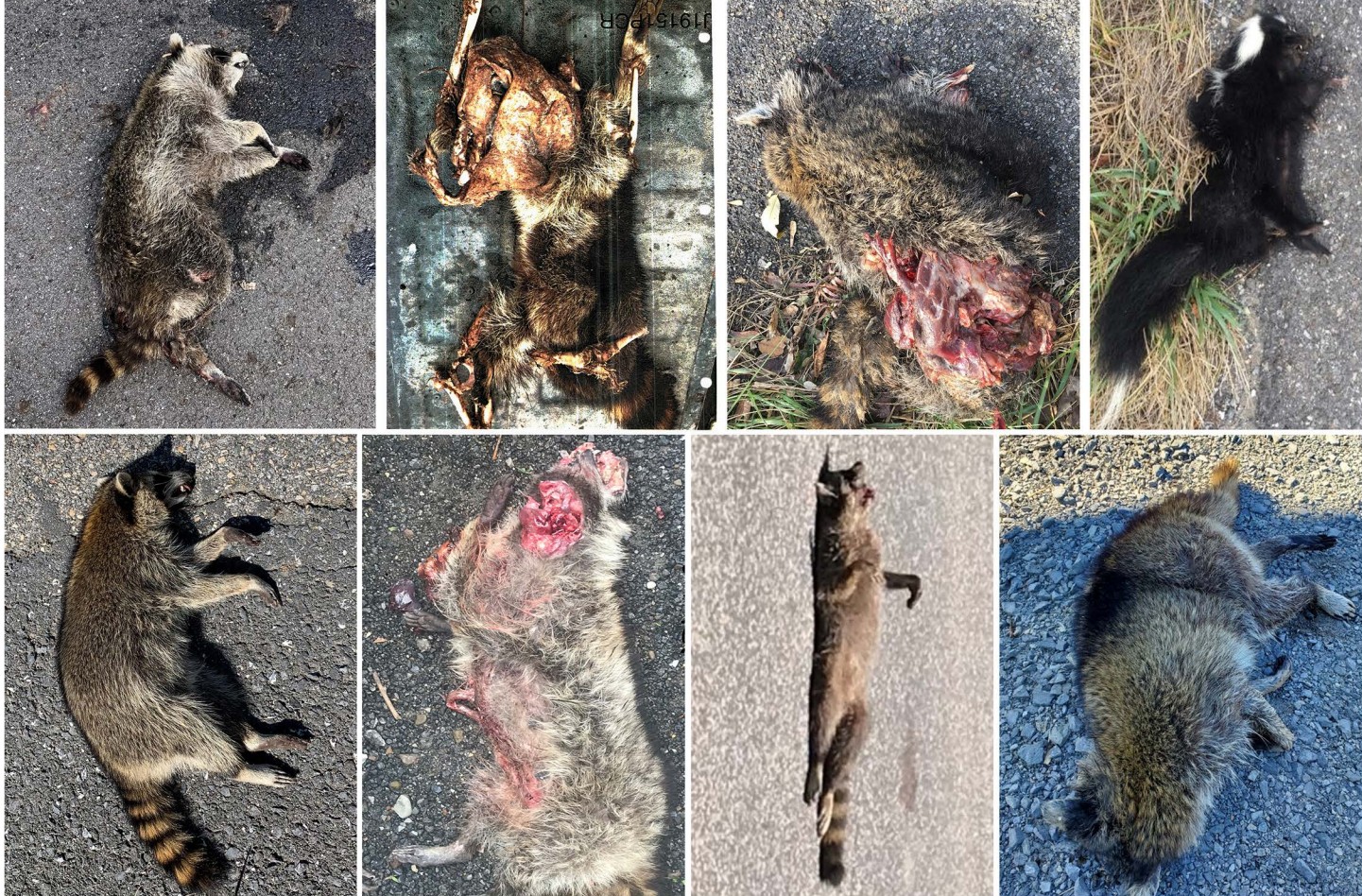

**Fig 2. Photos of roadkills testing positive for rabies using the LN34 real-time RT-PCR test.** Sample order is the same as Table 2, from top left to bottom right.

samples may not be an ideal approach. On the other hand, none of the positives had skull condition described as flattened or "no recognizable structures". Most samples that produced positive or negative results were collected from intact, fractured or severely fractured skulls (Fig 3). For any skull condition, ≥ 50% of samples produced indeterminate results, but higher rates of indeterminate results were observed for flattened skulls (80.0%) and roadkills with no recognizable structures (82.4%). Most samples producing positive or negative results were in the field for an estimated <1 or 1 – 3 days (Fig 3). Likewise, higher proportions of samples with <1 or 1 – 3 estimated days in the field produced positive or negative results (40.4% and 39.7%) than samples with >3 estimated days in the field (17.3%; Fig 3).

Based on this observed association of sample quality with likelihood of producing an indeterminate result, we created a multivariable logistic regression model to predict the testability of samples encountered during ERS activities, referred to as the sample degradation score. Samples characterized with a definitive positive or negative result were significantly associated with brain condition, skull condition, previous day's temperature, and the temperature at time of sample collection (Fig 4) with the equation:

$$Odds(Indeterminate) = -3.2152 + 0.31^*(Skull\ Condition) - 0.039^*(Previous\ Temperature)$$
$$+ 0.040^*(Days) + 0.91^*(Brain\ Condition)$$

PLOS Neglected Tropical Diseases

**Table 3. Rabies case detection rates for roadkill samples tested by LN34 by sample collection location relative to ORV zone and sample quality. Location is qualitative relative to the location of the 2018 ORV zone. Case detection rate is the percent positive out of total tested. P-values represent results of Mid-P exact test for the following comparisons of case detection rate: (top) LN34 Poor Quality Samples vs LN34 Good Quality Samples and (bottom) LN34 All Tested vs USDA ERS DRIT case detection rate for the same location relative to the ORV zone.**

| Location | Test Method | Ind | Negative | Positive | Total Tested | Case Detection Rate | P-value |
|---|---|---|---|---|---|---|---|
| **EAST** | LN34 - Poor Quality | 68 | 28 | 5 | 101 | 5.0% | 0.78 |
| | LN34 - Good Quality | 7 | 8 | 1 | 16 | 6.3% | |
| | **LN34 - All Tested** | **78** | **36** | **6** | **120** | **5.0%** | **0.03** |
| | DRIT - Routine Testing | 1 | 2,820 | 51 | 2,872 | 1.8% | |
| **ORV** | LN34 - Poor Quality | 89 | 20 | 2 | 111 | 1.8% | 0.88 |
| | LN34 - Good Quality | 1 | 6 | 0 | 7 | 0.0% | |
| | **LN34 - All Tested** | **93** | **29** | **2** | **124** | **1.6%** | **0.37** |
| | DRIT - Routine Testing | 0 | 3,273 | 105 | 3,378 | 3.1% | |
| **WEST** | LN34 - Poor Quality | 9 | 4 | 0 | 13 | 0.0% | Na |
| | LN34 - Good Quality | 2 | 1 | 0 | 3 | 0.0% | |
| | **LN34 - All Tested** | **12** | **5** | **0** | **17** | **0.0%** | **0.96** |
| | DRIT - Routine Testing | 1 | 4,321 | 10 | 4,332 | 0.2% | |

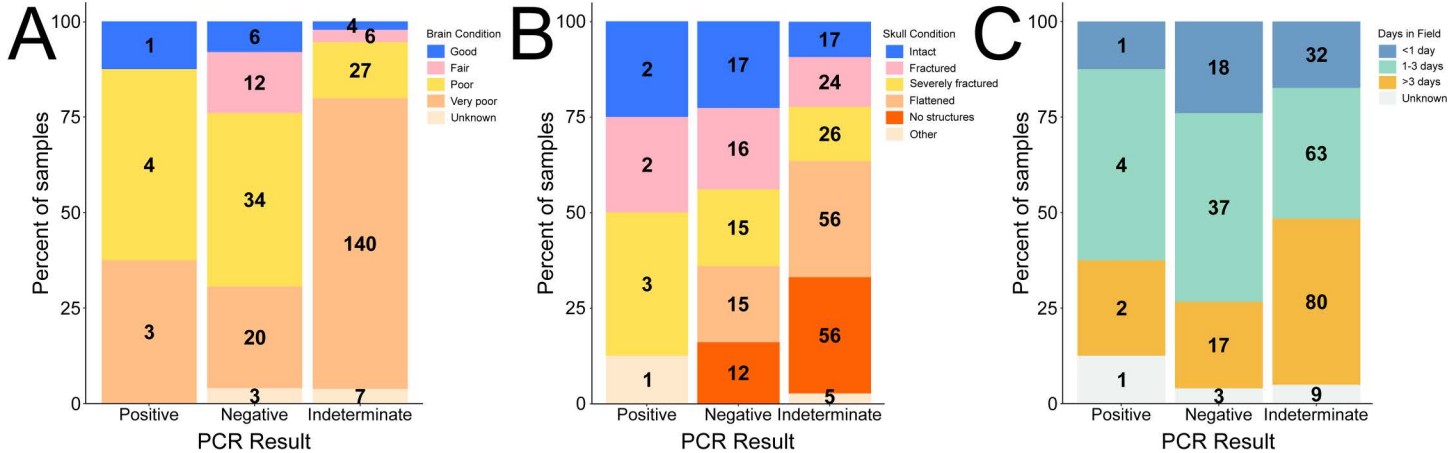

**Fig 3. Brain condition (A), skull condition (B), and estimated number of days in the field (C) for 267 roadkill samples tested for rabies using the LN34 real-time RT-PCR test.** Percent of positive, negative and indeterminate samples are shown by the size of colored bars and correspond to the y-axis. Number of samples per condition/result is written on bars. Brain condition, skull condition, and days in field were all chosen from a list of limited responses (see Methods). Samples from AZ were excluded (n = 32).

Seven samples were removed from the model, as they lacked temperature data. The multivariable logistic regression model showed good fit, with an AIC value of -276. Worsening skull condition, increasing temperature, and increasing number of days after presumed death were all associated with higher probabilities that the sample would yield indeterminate results and provide no diagnostic or surveillance benefit. Each of these conditions contribute to sample degradation and the two reasons for indeterminate results (1) inability to identify whether the sample collected contained brain tissue and (2) total degradation of RNA from the sample.

Model-derived probabilities that the sample would produce an indeterminate LN34 test result (e.g., the sample degradation score) were calculated for 267 study samples that had the model-required data elements. No ERS roadkill samples were expected to have sample degradation scores <0.10 (e.g., highest quality samples); therefore, no testing or case

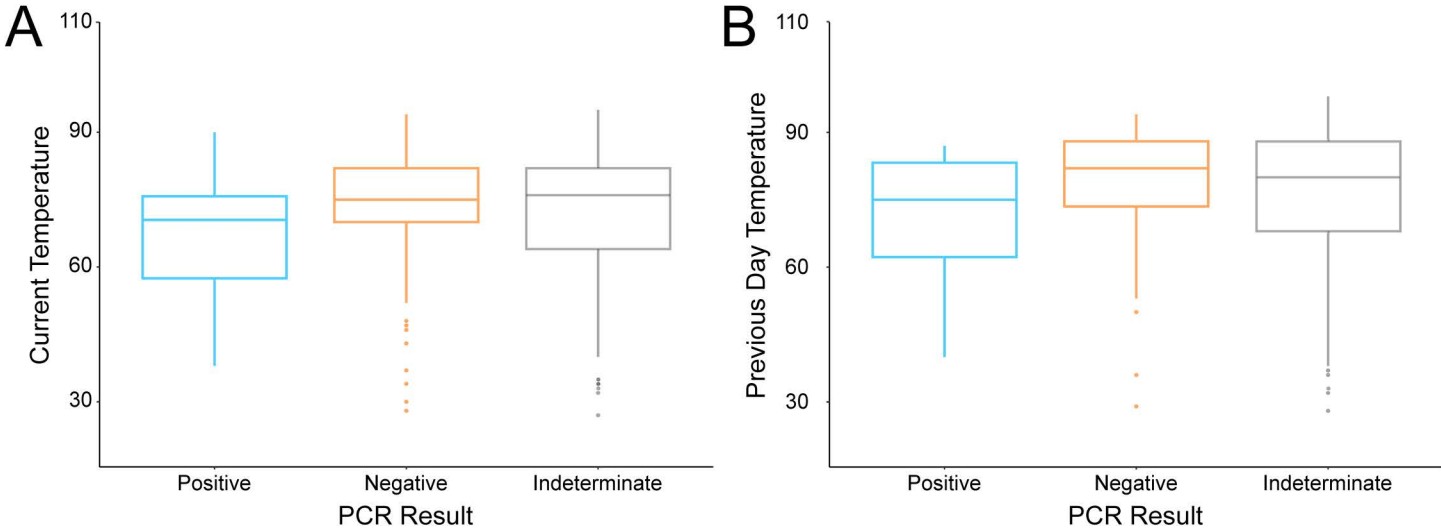

**Fig 4. Current (A) and previous day (B) temperature in the field for 267 roadkill samples tested for rabies using the LN34 real-time RT-PCR test.** Temperature was recorded in °F. Boxplots show median plus 25% and 75% quartiles. Whiskers show largest observation less than or equal to 1.5 * interquartile range. Outliers are shown as dots. Samples with no temperature data (n = 12 for A, n = 13 for B) were excluded. Samples from AZ were excluded (n = 32).

detections would occur. Only 1 of 8 positives samples in this study had a sample degradation score <0.40; the median degradation score for roadkill positive samples was 0.60 and one positive sample had a degradation score of 0.94. If a degradation score threshold of > 0.70 were used as a cutoff to collect and test, over half of samples collected would be expected to produce indeterminate results. The cost per sample with an informative result (positive or negative) rose dramatically at a degradation score >0.65, due to the high proportion of indeterminate results found at degradation score above this value (Fig 5). Testing all roadkill ERS samples encountered would result in a 2.5-fold increase in the cost per positive/negative result, when compared to programs that would use a sample degradation score threshold <0.60.

## Discussion

This study identified eight rabid animals that would likely not have been identified otherwise in areas of high interest for wildlife rabies management. These findings support the use of real-time RT-PCR for rabies surveillance using samples that would typically be unsuitable for testing by DFA and DRIT. We observed a 2.7% average positivity rate (CDR), 3.0% in states with RRVV, with rates from individual states ranging from 0% (AL, AZ, NC, OH, TN) to 10.7% (WV). This CDR reported is in line with RRVV CDRs in the areas, indicating that the LN34 assay may be a good technique for detecting rabies among roadkill samples. Previous reports of rabies CDRs among roadkill samples are limited but have generally reported rabies positivity <1% using DRIT or DFA [6,33] on higher quality samples from within and to the west of ORV zones, where RRVV has low prevalence or is absent [8,15]. In the current study, low quality samples were strategically collected from areas within and to the east of ORV zones (i.e., where RRVV is enzootic and a higher positivity rate would be expected). The positivity rate observed using LN34 in this study was more similar to overall positivity rates detected by ERS surveillance, when calculated separately for east of ORV zone (enzootic), within ORV zone or west of ORV zone (RRVV not present), even though sampling during this roadkill study was considerably different than typical ERS surveillance conducted by USDA-WS.

Roadkill surveys have been used to determine wildlife population trends [34–36], monitor diseases other than rabies [37], and perform population genomic studies [38]. In this study, roadkill surveillance for rabies virus detection was

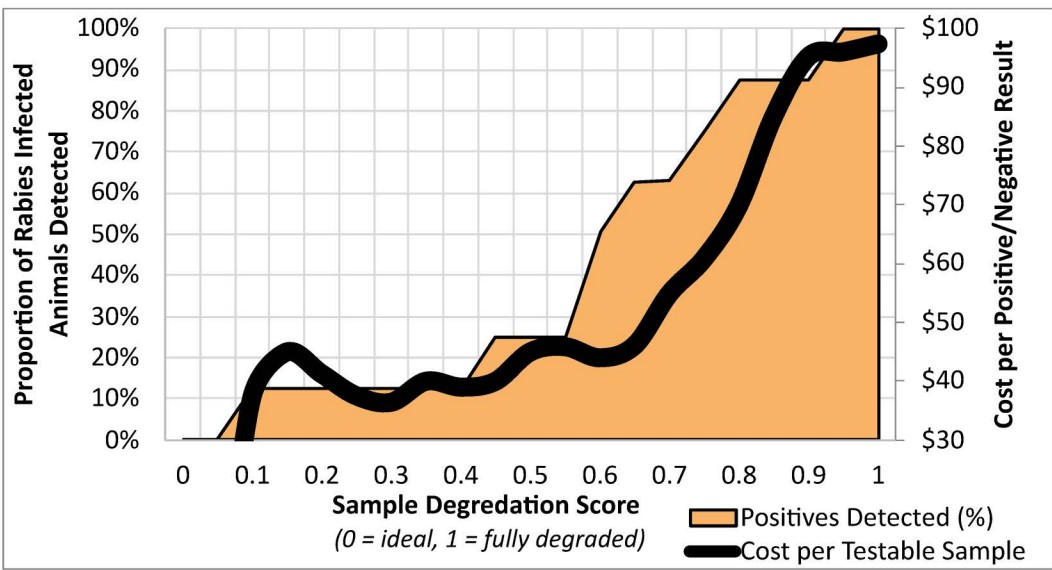

**Fig 5. Cost-benefit analysis comparing proportion of percent of rabies infected animals predicted to be detected (left axis, black line) and cost per informative result (positive/negative result, right axis, orange graph) at different sample degradation score thresholds (x-axis).**

performed opportunistically in some states and using formal, standardized roadkill surveys in other states, per the standard survey method for that state. Several positive rabies cases were identified from samples field biologists would likely drive by because of their poor sample condition. By implementing PCR for poor condition specimens and expanding the definition of acceptable sample conditions for testing, the number of samples tested can be increased in priority areas.

The high sensitivity of PCR-based methods can increase risk for false positive results caused by cross contamination. This is of particular concern for samples collected and processed in the field. In this study, we performed investigations to determine the risk of cross contamination for each positive result. We only identified one incident of likely cross contamination in which a disinfected scalpel was re-used to collect samples from two different animals, one of which produced a strong positive rabies PCR result. Best practices to avoid cross contamination include use of disposable, single-use supplies and gloves for each unique animal. This best practice is true, regardless of the diagnostic method (DRIT, DFA, or PCR), but may have more profound impacts on interpreting results when using PCR-based diagnostic methods.

In general, samples will be more likely to yield a definitive result (positive or negative) when they are in relatively good condition, temperatures are relatively low, and the sample is relatively fresh. However, well preserved, fresh tissue can also be tested by DFA or DRIT. We did not observe a difference in rabies positivity rate by LN34 real-time RT-PCR between poor and good quality samples; however, sample size was limited and the "good quality" designation for road-kills may not correspond to good quality for non-roadkill samples (i.e., in a clinical diagnostic case). Furthermore, the qualitative descriptions of roadkill are somewhat subjective and may differ from person to person depending on changing environmental conditions and their experience. In some cases, roadkill condition may not correspond to the brain sample condition. While the Wildlife Services biologists and technicians conducting sample collection for this study generally had extensive experience collecting and sampling roadkill, providing guidance and training in roadkill sample collection is critical for safety and standardization of roadkill testing.

It is of interest to identify the best type of tissue to collect from roadkills for rabies testing. In this study, brain, samples taken from inside the skull or skull fragments were most likely to produce a positive result. Three of the eight positive cases identified in this study had very poor condition and no identifiable brain tissue, so excluding very poor samples may result in missed

cases. While these samples did not contain recognizable brain tissue, they were collected from what remained of the head/cranial cavity and most likely contained very poor condition brain or nervous system tissues. No positive results were found for roadkill with skulls that were completely flattened or had no recognizable structures (n = 143), so roadkills with extreme damage to the head may not be worth testing. Lastly, tissue alternatives to brain or head/cranial material did not produce positive results in this study. Twenty-three skin samples were collected (S2 Table), but none produced a positive result. Previous studies have shown presence of rabies virus antigen and RNA in peripheral tissues including skin; however, rabies virus RNA and antigen are typically at highest levels in the brain, and brainstem is the best tissue for diagnosis if available [39–49].

No positives were identified among Arizona roadkills tested (n = 32). The Arizona samples were notable in their extreme desiccation. Future studies are needed to better understand the testability of poor-quality roadkill in Arizona including the stability of rabies virus RNA in a hot, arid climate. Published estimations of the expected rabies positivity rate in skunks or foxes in Arizona for roadkill are limited compared to rabies positivity rates around the RRVV ORV zone.

Surveys are conducted throughout the vaccine bait zones across many states with varying road types and traffic levels. Opportunistic sampling involves biologists picking up roadkill samples of target species often while driving to other work sites or to and from the office. Formal roadkill surveys involve standardized routes driven on a regular schedule.

Davis *et al.* (2023) [8] published a method for determining adequate sample sizes for determining freedom from RRVV, suggesting that as many as 4 – 15 samples may be required per 100 km$^2$, per year, to be confident in the absence of RRVV, depending on the landscape location of sampling in relation to management. Achieving a large sample size in defined areas of interest may require new approaches to ERS sampling strategies. The findings here suggest that the use of PCR diagnostic techniques can expand the quantity of samples collected during opportunistic and standardized roadkill ERS activities; however, expanding the sample size may also result in an increasing fraction of samples with indeterminate results. In this study, a negative PCR result required specific minimum quality controls in addition to absence of rabies virus RNA signal: (1) identifiable brain tissue in the sample collected and (2) no evidence of total RNA degradation. The ability to positively identify and collect brain tissue becomes increasing difficult with poor condition samples and from specimen with flattened or damaged skulls. To be conservative, samples with no identifiable brain were treated as non-brain samples since they may contain tissues where rabies virus is not reliably found during infection (muscle, bone, blood). Likewise, lack of lyssavirus or rabies virus RNA in a tissue where all RNA has been degraded is non-informative, and such a sample should not be considered negative.

The model presented here can be used to inform sample collection criteria for PCR testing of poor-quality samples based on sample degradation scores. Depending on the study design, it may be critical to detect all rabies positive samples, so limiting sample collection based on degradation score may not make sense. Similarly, small scale projects may be able to tolerate higher costs per positive/negative result. However, in the case of estimating rabies CDRs when there are abundant samples, using a sample degradation score cutoff could minimize testing of samples that are most likely to provide an indeterminate result. Depending on the indications for conducting roadkill testing, higher threshold values to determine if a sample should be collected may be used to improve sample sizes, despite the greater cost per testable sample. In the absence of situation-specific context(s), a sample testability threshold of 70% would likely maximize the CDR, while maintaining reasonable diagnostic testing costs for ERS.

LN34 testing in this study was performed in a laboratory using a conventional real-time PCR platform, biosafety cabinet, centrifuge, bead beater and other laboratory equipment. Field-based testing and point-of-care tests may be ideal for field biologists in some situations where rapid results are needed or when resources may not be available to perform diagnosis using large or expensive equipment. Point-of-care real-time RT-PCR tests are being developed and validated for use for viruses [50–53] including rabies virus [54–56]. Both PCR and antigenic [57–59] point-of-care tests for rabies hold a lot of promise to improve surveillance in field, remote and resource-limited settings.

## Supporting information

**S1 Table.  Sample details for roadkill samples tested in this study.**
(XLSX)

**S2 Table. Detailed PCR test results and sample details.**
(XLSX)

**S3 Table. Cost estimate for consumable supplies and reagents for RNA extraction and LN34 real-time RT-PCR testing.**
(XLSX)

**S4 Table. Details of re-extraction and re-testing of two samples producing LN34 Ct > 35 in initial testing.**
(XLSX)

**S1 Text. Field protocol for animal sample collection for RT-PCR Testing.**
(DOCX)

## Acknowledgments

We thank Lillian Orciari, Pamela Yager, Jesse Bonwitt, Cassandra Boutelle, Amy Gilbert, and USDA-WS and partner field employees in all states where samples were collected. Use of trade names and commercial sources is for identification only and does not imply endorsement by the Centers for Disease Control and Prevention, the U.S. Department of Health and Human Services, or the authors' affiliated institutions. The conclusions, findings, and opinions expressed by authors do not necessarily reflect the official position of the U.S. Department of Health and Human Services, the Centers for Disease Control and Prevention, or the authors' affiliated institutions.

## Author contributions

**Conceptualization:** Crystal M Gigante, Jordona D. Kirby, Kathleen M. Nelson, Ryan Wallace, Yu Li, Richard B. Chipman.

**Data curation:** Crystal M Gigante, Claire Hartloge, Rene Edgar Condori, Lauren Hovis.

**Formal analysis:** Crystal M Gigante, Rene Edgar Condori, Jordona D. Kirby, Lauren Hovis, Kathleen M. Nelson, Ryan Wallace, Richard B. Chipman.

**Funding acquisition:** Ryan Wallace, Richard B. Chipman.

**Investigation:** Crystal M Gigante, Claire Hartloge, Rene Edgar Condori, Lauren Hovis, Ryan Wallace.

**Methodology:** Crystal M Gigante.

**Project administration:** Jordona D. Kirby, Kathleen M. Nelson, Yu Li, Richard B. Chipman.

**Resources:** Richard B. Chipman.

**Supervision:** Richard B. Chipman.

**Validation:** Crystal M Gigante.

**Visualization:** Crystal M Gigante.

**Writing – original draft:** Crystal M Gigante, Ryan Wallace.

**Writing – review & editing:** Crystal M Gigante, Claire Hartloge, Rene Edgar Condori, Jordona D. Kirby, Kathleen M. Nelson, Ryan Wallace, Yu Li, Richard B. Chipman.

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
