## [Decision Letter · Decision Letter 0]

Dear Dr. Gigante,

Thank you very much for submitting your manuscript "Enhanced rabies surveillance in roadkill specimens by real-time RT-PCR" for consideration at PLOS Neglected Tropical Diseases. As with all papers reviewed by the journal, your manuscript was reviewed by members of the editorial board and by several independent reviewers. The reviewers appreciated the attention to an important topic. Based on the reviews, we are likely to accept this manuscript for publication, providing that you modify the manuscript according to the review recommendations.

Apologies for the delays in finishing the reviews of your manuscript. Both reviewers felt that your manuscript was a valuable contribution to the literature on rabies management. Both also had comments and suggestions for improving the manuscript. Please consider these comments, edit your manuscript accordingly and provide a point-by-point response to each of the reviewer comments. We look forward to seeing your revised manuscript and comments to reviewers.

Sincerely,

Richard A. Bowen

Academic Editor

Abdallah Samy

Section Editor

Apologies for the delays in finishing the reviews of your manuscript. Both reviewers felt that your manuscript was a valuable contribution to the literature on rabies management. Both also had comments and suggestions for improving the manuscript. Please consider these comments, edit your manuscript accordingly and provide a point-by-point response to each of the reviewer comments. We look forward to seeing your revised manuscript and comments to reviewers.

Reviewer's Responses to Questions

**Key Review Criteria Required for Acceptance?**

**Methods**

-Are the objectives of the study clearly articulated with a clear testable hypothesis stated?

-Is the study design appropriate to address the stated objectives?

-Is the population clearly described and appropriate for the hypothesis being tested?

-Is the sample size sufficient to ensure adequate power to address the hypothesis being tested?

-Were correct statistical analysis used to support conclusions?

-Are there concerns about ethical or regulatory requirements being met?

Reviewer #1: The objective of the study clearly aligned with stated hypothesis and the study design is appropriate to address the objective.the sample population is also appropriate and well described to test the hypothesis Sufficient sample size were also collected and analysed to support the conclusion of the study and I have no concern on the ethical and regulatory requirements.

Reviewer #2: The aim of this study is to evaluate the effectiveness of an enhanced rabies surveillance (ERS) strategy that tests roadkill samples by real-time RT-PCR (i.e., the LN34 assay). The authors point out that most data contributing to annual rabies surveillance in the U.S. is associated with passive surveillance programs in public health, in which the focus of laboratory testing for rabies is to manage potential rabies exposures in humans, pets and peridomestic animals. These data may lead to a biased understanding of the prevalence, distribution, and epizootiology of rabies within a given region because the public health focus generally leave gaps in our understanding of rabies circulation in wildlife in regions where fewer humans reside. The authors state that targeted active surveillance (i.e., ERS strategies) can expand our understanding of the prevalence and geographic distribution of wildlife rabies, thus improving rabies prevention and control activities, particularly in areas where active intervention programs (i.e., oral rabies vaccination) have been implemented to limit the spread of rabies. The objectives of the 4-year study were clearly stated; to evaluate the sensitivity of an enhanced sample collection/test method (i.e., the use of LN34 real-time RT-PCR test for rabies diagnosis in roadkill samples) to detect rabies cases in wildlife, to categorize samples based on likelihood of rabies detection using the LN34 RT-PCR method, and with that information, develop a multivariable logistic regression model to predict the likelihood of obtaining indeterminate results based on sample quality indicators such as skull and brain condition, temperature, and time in the field as well as perform a cost-benefit analysis of collecting and testing roadkill samples using different degradation score thresholds to provide guidance for best use of this enhanced surveillance strategy. The study design, which included collecting roadkill samples across ten states in known rabies enzootic areas and comparing this ERS method with data collected using traditional enhanced surveillance methods, is appropriate for the stated objectives. While the sample sizes were modest, they were sufficient to provide adequate power to support the hypotheses being tested. There are no concerns about this study not meeting ethical or regulatory requirements.

**Results**

-Does the analysis presented match the analysis plan?

-Are the results clearly and completely presented?

-Are the figures (Tables, Images) of sufficient quality for clarity?

Reviewer #1: The result of this study was clearly and completely presented and well illustrated in tables and figures.

Reviewer #2: The 4-year study focused on enhanced sampling of suspect rabid animals in areas of high interest for wildlife rabies management by collecting roadkill specimens that would otherwise be inappropriate for testing by traditional, gold standard surveillance tests (DFA and DRIT). These poor-quality samples were tested by real-time RT-PCR and case detection rates by this method were compared with enhanced rabies surveillance data reported to the National Rabies Surveillance System during two years of that same time period. While the sample size was modest, the study was able to support the use of real-time RT-PCR for rabies surveillance on samples that would be unsatisfactory for testing by traditional methods. Further, sample sizes were sufficient to provide adequate power to determine that this ERS strategy significantly increased case detection rates in the eastern U.S. raccoon rabies virus variant (RRVV) oral rabies vaccination (ORV) zone. Additionally, while most PCR results on suboptimal roadkill samples were indeterminate, they were able to determine that most samples categorized as fair or good produced definitive (reportable and actionable) PCR results, and based on this observation, they were able to develop a multivariable logistic regression model to assess conditions associated with either indeterminate or definitive PCR results and predict the testability of samples and a cost estimate for testing roadkill by RT-PCR. Model-derived sample degradation scores could be used to further refine sampling strategies to maximize probabilities that most samples would produce definitive results.

Data collected and analyses presented match the analysis plan and results, including tables and figures, were presented clearly.

**Conclusions**

-Are the conclusions supported by the data presented?

-Are the limitations of analysis clearly described?

-Do the authors discuss how these data can be helpful to advance our understanding of the topic under study?

-Is public health relevance addressed?

Reviewer #1: The conclusion of this study was well supported by the data well analysed and described. The authors have discussed how helpful it is to advance the understanding of rabies surveillance in road side killed animals. The rabies surveillance work and out put is more relevant to public health and well addressed the public health concern.

Reviewer #2: In this paper the authors explore the potential of ERS using RT-PCR methods to detect rabies virus in roadkill samples and find that this strategy can be a cost-effective way to complement public health surveillance of rabies and fill surveillance gaps, particularly in remote areas where surveillance is relatively poor. They demonstrate that RT-PCR (i.e., LN34) tests outperform gold standard DFA and DRIT tests when testing poor or deteriorated brain material. By adding RT-PCR testing of roadkill from the eastern U.S. raccoon rabies virus variant (RRVV) oral rabies vaccination (ORV) zone, they were able to significantly increase rabies case detection rates east of the ORV zone over CDRs obtained by traditional DRIT surveillance efforts.

Additionally, through development of a multivariable logistic regression model to predict the likelihood of obtaining indeterminate results based on sample quality indicators, the authors identify those samples that are more likely to lead to a definitive RT-PCR result (i.e., positive or negative versus indeterminate). This model may be a useful and practical tool to triage roadkill samples, thus yielding a higher percentage of definitive results (increasing case detection rates) while reducing the cost of the ERS effort.

**Editorial and Data Presentation Modifications?**

Reviewer #1: Accepted as it is with no minor edition and/or comment.

Reviewer #2: Overall, the authors have performed a well-designed study and provide very useful data on the potential of ERS using RT-PCR methods to detect rabies virus in roadkill samples to improve surveillance of wildlife rabies. The data are presented in a clear manner and the conclusions drawn about the tests are reasonable and not overstated. While this study warrants publication, there are a few minor modifications that would enhance clarity.

Methods

1. On Line 78 (also discussed on lines 326-328), the authors describe the states in which roadkill samples were collected for the study and mention that samples were collected through opportunistic or formal roadkill surveys, but do not define what the difference is between these survey approaches or detail if different survey strategies might affect or bias the numbers of roadkill samples collected or the potential for detecting positive cases (e.g., were the geographic regions that were targeted the most likely regions to detect wildlife rabies under both survey approaches)?

2. Line 89-90: The LN34 protocol in this study diverges from the published diagnostic LN34 protocol by reducing from triplicate to duplicate testing of the LN34 target and singlicate for the internal beta-actin control. From the validation studies of the diagnostic LN34 protocol, do the authors have data on the impact of reducing replicates on accurate detection of rabies virus RNA or beta actin in a run? If there is reduced sensitivity or accuracy in detections when reducing replicates, it would be helpful to state what is known.

3. Lines 100-101 and Table 3: For evaluating case detection rates, comparisons are made between LN34 Poor Quality Samples vs LN34 Good Quality Samples, but it is unclear how that determination was made. In evaluating brain condition, values of good, fair, poor, very poor are options. Was the “Good” vs “Poor” determination based only on brain condition or on a combination of brain condition and other factors (e.g., skull condition, temp, estimated # days in field)? If based upon brain condition, is “Good” = good and fair samples and "Poor" = poor and very poor?

4. Line 124 and CDC Datasheet: How does one estimate “Days in the Field” when one first encounters a roadkill specimen? Are there any guides to make that determination somewhat objective? If so, it would be helpful to describe.

5. Lines 126-127 and CDC Datasheet: Assessing some of the parameters that were used to describe the condition of samples collected in the field inevitably involves subjective judgement; however, it is not clear if there was any training or documentation provided to field collectors to attempt to standardize categorization of parameters that were checked on the “CDC datasheet for RT-PCR Roadkill Study with USDA” form and that were used to generate statistics for the study. For example, determination of brain condition as good, fair, poor, and very poor in S1 Table did not necessarily match with the physical description of the samples collected. There appeared to be similar descriptions that in some cases were categorized as good and in other cases as fair or poor. It would be very helpful to discuss any attempts to try to standardize those categorizations or discuss the limitations of not being able to standardize.

Results

6. Line 219, it is mentioned that there was no significant difference when comparing CDRs for samples designated as poor or very poor to “either good or very good quality”. Were there any samples designated as very good? That term is not listed under the criteria for modeling or in the supplemental tables.

7. Lines 196-204: In the descriptions of the three positive samples that contained no recognizable brain tissue, particularly the first sample (i.e., leathery brown tissue) was there further investigation with the field collectors to obtain more specific details about what they targeted for collection? Despite that the samples “contained no recognizable brain tissue”, the Ct values were reasonably good, so was it thought that the samples were in fact extremely poor-quality brain tissue or it is thought that other tissues might yield positive results in the LN34 assay? Are there tissues studies that suggest alternative sample types might be useful for sampling?

**Summary and General Comments**

Reviewer #1: I think the surveillance work is impactful if extrapolated to other regions like Africa where road side killings of small and large wild animals around the National park areas is regularly occur. This information would also help to further investigate other wild life pathogens of zoonotic origin.

Reviewer #2: This manuscript is well organized and well written and provides strong evidence for their claims. The data support that RT-PCR (LN34) is more sensitive than DFA or DRIT in detecting rabies virus RNA in suboptimal (deteriorated or decomposed) brain samples and incorporation of RT-PCR testing of roadkill animals into ERS efforts can enhance case detection rates and offers practical guidance from cost-benefit analysis of collecting and testing roadkill samples using different degradation score thresholds. Overall, this study is an important contribution by proposing a viable strategy that complements current rabies surveillance methods by providing rabies surveillance data on important wildlife reservoirs in areas where passive public health surveillance data are missing and enhance wildlife surveillance in critical areas to support both regional and national rabies management programs.

PLOS authors have the option to publish the peer review history of their article (what does this mean? ). If published, this will include your full peer review and any attached files.

**Do you want your identity to be public for this peer review?** For information about this choice, including consent withdrawal, please see our Privacy Policy .

Reviewer #1: Yes: Asefa Deressa

Reviewer #2: No

Figure Files:

Data Requirements:

Reproducibility:

References

---

## [Decision Letter · Decision Letter 1]

Response to Reviewers
Revised Manuscript with Track Changes
Manuscript

We look forward to receiving your revised manuscript.

Shaden Kamhawi

co-Editor-in-Chief

Paul Brindley

co-Editor-in-Chief

**Additional Editor Comments :**
**Journal Requirements:**

1)  Please remove your responses to the Journal Requirements from your Cover Letter.

- TM on page: 5.

3) When completing the data availability statement of the submission form, you indicated that you will make your data available on acceptance. We strongly recommend all authors decide on a data sharing plan before acceptance, as the process can be lengthy and hold up publication timelines. Please note that, though access restrictions are acceptable now, your entire data will need to be made freely accessible if your manuscript is accepted for publication. This policy applies to all data except where public deposition would breach compliance with the protocol approved by your research ethics board. If you are unable to adhere to our open data policy, please kindly revise your statement to explain your reasoning and we will seek the editor's input on an exemption. Please be assured that, once you have provided your new statement, the assessment of your exemption will not hold up the peer review process.

**Reviewers' comments:**

**Key Review Criteria Required for Acceptance?**

**Methods**

-Are the objectives of the study clearly articulated with a clear testable hypothesis stated?

-Is the study design appropriate to address the stated objectives?

-Is the population clearly described and appropriate for the hypothesis being tested?

-Is the sample size sufficient to ensure adequate power to address the hypothesis being tested?

-Were correct statistical analysis used to support conclusions?

-Are there concerns about ethical or regulatory requirements being met?

Reviewer #1: The objective of the study design is well articulated and appropriately addressed it. The sample population is clearly described and the hypotheses being tested is appropriate as well. The sample size is also sufficient to ensure and address the hypotheses being tested.The correct statistical analysis was used.

Reviewer #2: See below Summary and General Comments.

**Results**

-Does the analysis presented match the analysis plan?

-Are the results clearly and completely presented?

-Are the figures (Tables, Images) of sufficient quality for clarity?

Reviewer #1: Results are clearly and completely presented matching the analysis plan.

Reviewer #2: See below Summary and General Comments.

**Conclusions**

-Are the conclusions supported by the data presented?

-Are the limitations of analysis clearly described?

-Do the authors discuss how these data can be helpful to advance our understanding of the topic under study?

-Is public health relevance addressed?

Reviewer #1: The conclusion of the study was supported by the data analyzed and presented. The authors have also discussed to advance the understanding of the study subject and topic.

Reviewer #2: See below Summary and General Comments.

**Editorial and Data Presentation Modifications?**

Reviewer #1: Accept as it is.

Reviewer #2: See below Summary and General Comments.

**Summary and General Comments**

Reviewer #1: Regardless of the adequate number of sample size the positive samples were with insignificant number. There is no duplication of publication. No concern of research ethics and publication ethics.

Reviewer #2: This manuscript is well organized and well written and provides strong evidence for their claims. The data support that RT-PCR (LN34) is more sensitive than DFA or DRIT in detecting rabies virus RNA in suboptimal (deteriorated or decomposed) brain samples and incorporation of RT-PCR testing of roadkill animals into ERS efforts can enhance case detection rates and offers practical guidance from cost-benefit analysis of collecting and testing roadkill samples using different degradation score thresholds. Overall, this study is an important contribution by proposing a viable strategy that complements current rabies surveillance methods by providing rabies surveillance data on important wildlife reservoirs in areas where passive public health surveillance data are missing and enhance wildlife surveillance in critical areas to support both regional and national rabies management programs.

Upon initial review, some minor modifications were recommended in the methods, results and discussion sections. The authors responded to all recommended modifications enhancing clarity and correcting minor errors. I have no further suggested edits or changes to the manuscript and recommend that it be accepted for publication.

PLOS authors have the option to publish the peer review history of their article (what does this mean? ). If published, this will include your full peer review and any attached files.

**Do you want your identity to be public for this peer review?** For information about this choice, including consent withdrawal, please see our Privacy Policy .

Reviewer #1: **Yes: ** Hundera Asefa Deressa

Reviewer #2: No

**Figure resubmission:****Reproducibility:** To enhance the reproducibility of your results, we recommend that authors of applicable studies deposit laboratory protocols in protocols.io, where a protocol can be assigned its own identifier (DOI) such that it can be cited independently in the future. Additionally, PLOS ONE offers an option to publish peer-reviewed clinical study protocols. Read more information on sharing protocols at https://plos.org/protocols?utm_medium=editorial-email&utm_source=authorletters&utm_campaign=protocols

---

## [Editor Report · Decision Letter 2]

Dear Dr. Gigante,

We are pleased to inform you that your manuscript 'Enhanced rabies surveillance in roadkill specimens by real-time RT-PCR' has been provisionally accepted for publication in PLOS Neglected Tropical Diseases.

Best regards,

Richard A. Bowen, DVM PhD

Academic Editor

Abdallah Samy

Section Editor

Shaden Kamhawi

co-Editor-in-Chief

Paul Brindley

co-Editor-in-Chief

---

## [Editor Report · Acceptance letter]

Dear Dr. Gigante,

We are delighted to inform you that your manuscript, "Enhanced rabies surveillance in roadkill specimens by real-time RT-PCR," has been formally accepted for publication in PLOS Neglected Tropical Diseases.

Best regards,

Shaden Kamhawi

co-Editor-in-Chief

Paul Brindley

co-Editor-in-Chief
